# Autocrine Neuromodulation and Network Activity Patterns in the Locus Coeruleus of Newborn Rat Slices

**DOI:** 10.3390/brainsci12040437

**Published:** 2022-03-25

**Authors:** Quinn Waselenchuk, Klaus Ballanyi

**Affiliations:** Department of Physiology, Faculty of Medicine & Dentistry, University of Alberta, Edmonton, AB T6G 2H7, Canada; qwaselen@ualberta.ca

**Keywords:** astrocytes, brain slices, calcium imaging, calcium wave, local field potential, locus coeruleus, noradrenaline, neonatal, oscillations, pattern transformation, rhythm generation, synchronization

## Abstract

Already in newborns, the locus coeruleus (LC) controls multiple brain functions and may have a complex organization as in adults. Our findings in newborn rat brain slices indicate that LC neurons (i) generate at ~1 Hz a ~0.3 s-lasting local field potential (LFP) comprising summated phase-locked single spike discharge, (ii) express intrinsic ‘pacemaker’ or ‘burster’ properties and (iii) receive solely excitatory or initially excitatory–secondary inhibitory inputs. μ-opioid or ɑ_2_ noradrenaline receptor agonists block LFP rhythm at 100–250 nM whereas slightly lower doses transform its bell-shaped pattern into slower crescendo-shaped multipeak bursts. GABA_A_ and glycine receptors hyperpolarize LC neurons to abolish rhythm which remains though unaffected by blocking them. Rhythm persists also during ionotropic glutamate receptor (iGluR) inhibition whereas <10 mV depolarization during iGluR agonists accelerates spiking to cause subtype-specific fast (spindle-shaped) LFP oscillations. Similar modest neuronal depolarization causing a cytosolic Ca^2+^ rise occurs (without effect on neighboring astrocytes) during LFP acceleration by CNQX activating a TARP-AMPA-type iGluR complex. In contrast, noradrenaline lowers neuronal Ca^2+^ baseline via ɑ_2_ receptors, but evokes an ɑ_1_ receptor-mediated ‘concentric’ astrocytic Ca^2+^ wave. In summary, the neonatal LC has a complex (possibly modular) organization to enable discharge pattern transformations that might facilitate discrete actions on target circuits.

## 1. Introduction

The locus coeruleus (LC) in the dorsal pons is the source for actions of its main neurotransmitter noradrenaline (NA) on most structures in the central nervous system (CNS). Consequently, the LC controls multiple behaviors such as arousal, sleep–wake cycle, breathing, memory, pain sensation, anxiety, and opioid (withdrawal) effects via activity-related NA release [1,2,3,4,5].

### 1.1. Connectivity of the Adult LC

The complex connectivity in the adult LC as described in this section is summarized in Figure 1. Regarding afferent inputs, e.g., the nucleus paragigantocellularis as well as the orbitofrontal and anterior cingulate cortices release glutamate onto LC neurons to activate all ionotropic glutamate receptor (iGluR) subtypes, i.e., receptors for α-amino-3-hydroxy-5-methyl-4-isoxazole propionic-acid (AMPAR), kainate (KAR), and N-methyl-D-aspartate (NMDAR). LC neurons also express postsynaptic receptors for neuropeptide transmitters, such as corticotropin-releasing factor or hypocretin/orexin, that are activated by inputs from the paraventricular nucleus and the posterior lateral hypothalamus, respectively. Moreover, serotonin receptors (5-HTR) are activated on LC neurons by inputs from the Raphe nuclei whereas they are inhibited by ventrolateral preoptic area neurons that release γ-aminobutyric acid (GABA) to act on GABA_A_ receptors (GABA_A_R). Importantly, interneurons within the LC can also release GABA to activate GABA_A_R on neighboring cells and local release of opioids or NA (from proximal LC neuron collaterals) has further modulating effects on LC activity [6,7,8]. While opioid actions are presumably inhibitory, NA can either activate typically inhibitory ɑ_2_ (auto)receptors (ɑ_2_R) or excitatory ɑ_1_ (auto)receptors (ɑ_1_R) on LC neurons and, the latter, also on neighboring astrocytes. LC astrocytes likely also express other neurotransmitter and neuromodulator receptors which mediate neuron-glia (‘neural’) network interactions that support behaviors such as sleep, attention, or breathing [9,10]. The latter examples indicate that the LC neural network is under extensive autocrine neuromodulatory control.

In response to the concerted activities from afferent structures, LC neurons discharge Na^+^ action potential ‘spikes’ that promote release of NA at axon varicosities in the target areas plus presumably a co-transmitter such as the neuropeptides galanin and neuropeptide-Y [3]. This LC activity modulates the patterns of often spontaneous activities in the target circuits like different cortical areas, the thalamus, or the cerebellum. Moreover, activities in these circuits can change during various brain states such as sleep or wakefulness, pain sensation, or attention, already without input from the LC. Altogether this means that the LC serves as an interface. On the one hand, it receives complex patterns of synaptic inputs from various sources to cause release of different types of neurotransmitters acting on several receptor subtypes. On the other hand, this evokes LC neuron discharge and/or alters their spontaneous spiking to subsequently modulate the activity in target circuits as well as within the LC for autocrine neuromodulation. 

### 1.2. Modular Organization of Adult LC

It is not known how the small and compact LC, consisting in each of its bilaterally-organized aspects of only ~1600 neurons in rats (Figure 2) and ~20,000 in humans [3,12,13], can control many different brain activities and behaviors. A recent review of this topic [14] stated that it was thought until recently that the evolutionarily ancient LC sends a rhythmic and global NA pulse to the neuraxis, similar to rhythmic systemic blood distribution by the heart. Accordingly, LC neurons and heart myocytes would share electrophysiological membrane properties like gap junction-mediated electrical coupling or intrinsic ‘pacemaker’-like ion conductances mediating spontaneous spiking to generate coordinated activity for pulsed NA release and blood supply, respectively. The authors then point out that in the past decade evidence has steadily increased that the LC contains anatomically and functionally distinct modules. As examples from various studies [3,5,15,16,17,18,19], LC neurons show a topographical organization regarding: (i) morphology that shows a preferentially ‘multipolar’ shape of the soma and primary dendrites in the ventral LC aspect contrary to location of simpler ‘fusiform’ shaped neurons in the dorsal part, (ii) release of a distinct co-neurotransmitter such as neuropeptide-Y or galanin in addition to NA, (iii) expression of neurotransmitter receptors such as ɑ_1_R or ɑ_2_R, (iv) axonal projection areas as neurons innervating the hippocampus are preferentially located in the anterio-dorsal LC region and those projecting to the cerebellum in the ventral aspect, (v) electrophysiological properties as neurons innervating distinct cortical areas differ in their firing rate and spike after-hyperpolarization amplitude. There is also evidence for a pontospinal-projecting module that is comprised of ventrally-located LC neurons with a shorter spike and smaller spike after-hyperpolarization compared to the LC core whereas faster spiking with enhanced adaptation is seen in dorsomedially-located small GABAergic neurons. In addition to such modular organization of the LC, its network activity is complex rather than occurring mainly in pulsed fashion. Specifically, there is evidence for an ‘ensemble code’ in the LC and spiking does not occur as a synchronous population event. Instead, LC neurons are capable of producing different outputs with varying extent of synchrony between them which may be correlated to behavioral states such as vigilance. Altogether, these considerations point out that the adult LC is a complex network with a modular organization needed for its diverse control functions.

### 1.3. Current Knowledge of Neonatal LC Network Properties

In rats, axons of LC neurons project to diverse target areas already at 12–14 days of gestation [23,24]. This implies that the neonatal LC is functional and plays an important role in controlling development of other brain networks in the perinatal time period [6,21,25,26,27]. Due to its small size there is very sparse information on electrophysiological and pharmacological LC properties in intact newborn animals [6,28,29]. Consequently, such knowledge is primarily based on findings from in vitro models. Studies on newborn rat brain slices have established that, at this age, the LC comprises a ‘simple’ spontaneously active neural network that serves as a model for spike synchronization [13,27]. Specifically, neonatal LC neurons are electrically coupled via gap junctions and generate, via an endogenous pacemaker mechanism, synchronized Ca^2+^-dependent subthreshold oscillations (STOs) of their membrane potential (V_m_) at a rate of ~1 Hz that lead to discharge of typically a single Na^+^ spike at their peak [26,30,31,32,33] (Figure 3). During the first 2–3 postnatal weeks, the rate of such tonic spiking increases slightly (to ~3 Hz) and the extent of synchronization appears to weaken progressively [26,27,34]. The latter studies proposed that decreased spike synchronization is the consequence of diminished gap junction expression.

While the aforementioned findings were obtained using neonatal rat slices, many other LC neuron properties were analyzed in slices from either juvenile or adult rodents. For example, to study afferent inputs, V_m_ recording from a single neuron was combined with electrical stimulation of neighboring slice areas [38]. This showed that LC neurons express different types of ion channels mediating a postsynaptic potential (PSP), specifically (i) iGluR cation channels causing an excitatory postsynaptic potential (EPSP), (ii) anion channels associated with a GABA_A_R or glycine receptor (GlyR) mediating an inhibitory postsynaptic potential (IPSP), and also (iii) slow (post-activation) IPSPs caused by G protein receptor-coupled K^+^ (‘GIRK’) channels associated with ɑ_2_(auto)R that are activated by local release of NA from neighboring LC neurons for autocrine modulation.

In brain slices, it is not clear from which areas in the neuraxis stimulated axons originate. In contrast, using a newborn rat brainstem-spinal cord model it was shown that the LC receives rhythmic inputs from brainstem respiratory networks [21] (Figure 2). Specifically, >50% of recorded tonically active LC neurons show an inspiratory-related burst of iGluR-mediated EPSPs which accelerates their spiking. This excitation is followed by an ɑ_2_R-mediated hyperpolarization during which spiking is blocked. This report also showed that these LC neurons are chemosensitive as their spike rate increases in solution with a lower pH. It is possible that this chemosensitivity is fed back synaptically into the inspiratory network to adapt its activity to a change in pH in the brainstem [4]. Finally, they identified in that study ‘type-1′ and ‘type-2′ LC neurons that differ in STO regularity and amplitude (Figure 2). In summary, this work indicated that the neonatal LC is already a complex network with functionally different neuron types. Note that an interaction of the LC and respiratory neural circuits is also seen in adults as shown in an article of this Special Issue [39].

### 1.4. Aim of Present Study

In this ‘Perspective’ article, we present further evidence for a complex (possibly even modular) organization of the neonatal LC based on our recent findings in acutely isolated horizontal slices from newborn rats. Specifically, our experiments were done on horizontal brain slices from 0–7 day-old CD-001 (Sprague-Dawley) rats of either sex (Charles River Laboratory Inc., Wilmington, MA, USA). Slices were generated using the procedures described previously (34, 39, 40). The success rate of generating a slice that could be used for recording was >95%. In almost all cases one finding was obtained from one recording in one slice per rat. 

Contrary to more challenging in vivo studies, cells in such slices can be visualized while combining extracellular and intracellular electrophysiological recording and live cell imaging, e.g., of changes in their free cytosolic Ca^2+^ concentration (Ca_i_), during quantitative pharmacological analyses. In this regard, we were the first to demonstrate that the LC generates a local field potential (LFP) in these slices [22] (Figure 2) (note that this LFP can principally also be termed ‘multi-unit activity’ [40,41,42]). In this report, we show that the same slices also contain networks in the entorhinal cortex and hippocampus which generate rhythmic LFPs different from that in the LC regarding both their rates and mechanisms (see also [43]) (Figure 2). In a more recent study [36], we combined LFP recording of LC neuron population activity with single neuron recording to show that their spiking is not synchronous and is, instead, ‘phase-locked’ to the LFP. In a third study [35], we used LFP plus whole-cell patch-clamp V_m_ recording as well as multiphoton Ca_i_ imaging in populations of LC neurons and neighboring astrocytes during LFP acceleration evoked by activation of AMPARs forming a complex with supplementary proteins. Here, we put these results in context with our current findings based on consistent effects in at least five experiments. In summary, our study indicates that the neonatal LC shares various cellular and pharmacological properties and has a similarly complex (possibly modular) organization, as in adults. In the discussion, we show perspectives for future directions of research on autocrine neuromodulation in the neonatal and adult LC.

## 2. Results

We firstly deal with intrinsic properties of the LC network and then state a lack of necessity of iGluR-mediated excitation and receptor-coupled anion channel-mediated inhibition for generation of LFP rhythm. We then show that iGluR agonists accelerate neuronal spiking and transform the LFP pattern. Next, we demonstrate that high doses of the µ opioid receptor (µR) agonists morphine and [D-Ala^2^, *N*-MePhe^4^, Gly-ol]-enkephalin (DAMGO) and the ɑ_2_R agonist clonidine abolish rhythm, whereas low doses of these agents slow it with occurrence of an often crescendo-like multipeak LFP pattern. This is followed by pointing out different NA effects on LC neurons and astrocytes as revealed by population Ca_i_ imaging. It is also noted under which conditions rhythmic activities and different types of LC neurons can be detected with population Ca_i_ imaging. Finally, we refer to a neuron–astrocyte interaction involving lactate as a gliotransmitter.

### 2.1. Anatomical and Intrinsic LC Properties

Neonatal rat LC neurons have a rather uniform round to fusiform shape with a soma diameter of 20–30 μm from which 2–5 processes originate in a confocal plane (Figure 3). These neurons are intermingled with fewer astrocytes with a soma diameter of ~10 μm whereas similarly sized astrocytes surrounding the LC form a more dense network (Figure 3). The largest diameter of the spindle-shaped neonatal rat LC in the horizontal plane is ~300 μm and it extends in the dorso-ventral plane by ~1000 μm [22,35,44,45]. We found in 400 μm thick acutely isolated horizontal slices that positioning of a suction electrode with an outer tip diameter of 40–60 μm at slice surface within the LC reveals a rhythmic LFP at a rate of ~1 Hz that is due to summation of mostly single spikes in 3–10 neurons located close to the electrode [36]. The LFP has a similar shape and amplitude when the electrode is positioned either in the center or more peripheral LC areas indicating that it is quite uniform within the nucleus. At a distance greater than ~50 μm outside the LC neuron somata area, no robust LFP is detectable. Moreover, the LFPs of the ipsilateral and contralateral LC aspects do not show a temporal correlation. The LFP signal generally has a very good signal-to-noise ratio and is stable for up to 24 h, therefore making it well suited for complex functional and pharmacological analyses of neonatal LC network properties. The duration of a single, mostly bell-shaped, burst (lasting ~0.3 s) is particularly evident in the ‘integrated’ form of the signal. This duration already indicates that the LFP does not represent the summation of synchronous Na^+^ spikes that span 1–3 ms in (neonatal) LC neurons [31,46,47,48] (Figure 3 and Figure 4). By combining LFP recording with patch electrode recording of single neuron spiking, we showed that each cell discharges (with a jitter of 20–100 ms) preferentially at a particular time point of the population burst [36] (Figure 3). We concluded in that study that neonatal LC neuron discharge is not synchronous, but rather ‘phase-locked’ to the network response. 

Regarding other intrinsic LC network properties, we demonstrated that the LFP is reversibly abolished by the voltage-gated Na^+^ channel blocker tetrodotoxin (TTX), indicating that it reflects neuronal spiking rather than STOs [36]. However, we found that TTX also abolishes STOs in most LC neurons [35]. The presence of the ‘persistent’ subtype of Na^+^ channels (mediating neuronal bursting) in the newborn rat LC is indicated by our finding in slices that the blocker riluzole abolishes the LFP [22] (Figure 3). 

With respect to the connectivity of neonatal LC neurons, it has been proposed that STOs and associated spiking are synchronous due to their electrical coupling by gap junctions comprising connexin-26, -32, and -43 subunits [13,44,51] (Figure 3). Another study instead proposed that neuronal coupling occurs only via connexin-36 [52]. Accordingly, 100 μM of the gap junction inhibitor carbenoxolone abolished STOs and associated rhythmic hyperpolarizations in LC astrocytes [34,51] while we found that the connexin-32 blocker mefloquine also blocks LFP rhythm [36] (Figure 3). In our hands, carbenoxolone application for ≥5 min only transiently perturbs LFP rhythm at 100 μM or more while mefloquine also depresses neuronal properties. It thus appears that more selective gap junction blockers are needed to identify the specific functional role of gap junctions in the neonatal LC. 

We also found that LC neurons differ regarding other intrinsic conductances not involving neurotransmitter actions [20]. Specifically, three distinct spike patterns are seen in response to changing their ‘holding’ V_m_ via injection of constant current through the patch electrode from their ‘resting’ V_m_ ranging from −35 to −55 mV [21,31,35] (Figure 2
Figure 4, Figure 5, Figure 6 and Figure 7). As exemplified in Figure 4, 52% of 29 recorded neurons respond to such depolarization with a steady increase of the rate of their very regular spiking. Another 28% of LC neurons discharge spike trains (‘bursts’) in response to depolarizing their V_m_, and with increasing sustained depolarization by up to 25 mV, these trains consist of 2 to 15 spikes/s that are interrupted by progressively shorter silent periods of ~1 to 0.1 s. The remaining 21% of cells respond to depolarization with modestly accelerated spiking which is quite irregular, a feature also seen often at resting V_m_.

As will be partly dealt with in more detail below, LFP rhythm is not substantially depressed in slices with robust rhythm by blockers of neurochemical synaptic processes involving AMPAR, KAR, NMDAR, GABA_A_R, GlyR, μR, (auto) ɑ_1_R, ɑ_2_R, or β NA receptors (βR). It can be concluded that these receptors are not required for generating this network rhythm.

### 2.2. Role of iGluR in Neonatal LC Rhythm

Various neural network rhythms in newborn and adult mammals depend on iGluR (for references, see Section 3 Discussion). For example, it was mentioned above that horizontal newborn rat brain slices contain spontaneously active networks in the hippocampus and entorhinal cortex in addition to that in the LC (Figure 2). The ‘classical’ AMPAR antagonist 6-cyano-7-nitroquinoxaline-2,3-dione (CNQX) [53] is sufficient to abolish the ‘early network oscillations’ in the entorhinal cortex [43] whereas combined blockade of both AMPAR and NMDAR is needed to abolish the hippocampal oscillations [54]. In the LC of such slices, we found firstly that the unselective non-competitive iGluR blocker kynurenic acid has no effect on LFP rhythm [35]. However, in the same study 25 μM CNQX accelerated LFP bursting in association with a <10 mV neuronal depolarization and a modest rise in neuronal Ca_i_ (see below). All effects were reversed by 25 μM the non-competitive AMPAR blocker GYKI-53655. We concluded from these findings that in neonatal LC neurons AMPAR form a functional complex with transmembrane AMPAR regulatory proteins (TARP) on which CNQX acts as a partial agonist [53,55,56,57]. The latter findings and the effects of iGluR agonists described in the following are summarized in Figure 5.

While generation of LC network rhythm does not depend on iGluR, AMPAR, KAR and NMDAR, agonists at these receptors have pronounced effects on LFP pattern [49,50]. Specifically, moderate doses of these iGluR agonists accelerate LFP rhythm from ~1 Hz to ~5 Hz. The faster LFP oscillations during AMPA and KA occur frequently without recovery of the (integrated) signal to baseline while showing a spindle-shaped waxing and waning of amplitude. At concentrations of 0.25–0.5 μM AMPA and 2.5 μM KA, these oscillations are very stable and similar to each other. During the LFP oscillations, V_m_ depolarizes by ~5 mV and spike rate increases by the same rate as LFP rhythm. While kynurenic acid blocks both types of LFP oscillations, GYKI-53655 selectively blocks the responses to AMPA whereas KA-evoked oscillations are selectively abolished by 25 μM UBP-302. NMDA evokes similar fast LFP oscillations at 25–50 μM that are selectively blocked by (2R)-amino-5-phosphonovaleric acid (APV). The oscillations are interrupted during sustained NMDA application after 1–5 min by a 1–2 s-lasting blockade of rhythm resulting in ‘oscillation trains’. In single LC neurons, the time courses of crescendo-shaped rhythmic sustained V_m_ depolarizations, intermittent hyperpolarizations, and spiking are closely related to the oscillation train LFP pattern (Figure 5).

The acceleration of rhythm by CNQX and all iGluR agonists is accompanied by a shortening of single burst duration, e.g., for CNQX by 26% [35]. This shortening indicates that the extent of spike synchronization is increased by the agonists. In the case of CNQX, though, cross-correlation analysis of the LFP peak with spiking in single neurons did not verify this assumption while we did find that the regularity of rhythm increased [35]. In contrast, NMDA increases the extent of synchronization while AMPA and KA show a trend in that respect. Regardless, all agonists increase the regularity of rhythm.

### 2.3. Role of Inhibition on Neonatal LC Rhythm

Ongoing synaptic inhibition via GABA_A_R or GlyR is needed for several types of brain rhythms in adults and newborns (for references, see Section 3 Discussion). In contrast, neonatal LC network rhythm is not affected by blockade of GABA_A_R or GlyR as exemplified in Figure 6, which also demonstrates that LFP rhythm is abolished by activation of these receptors with muscimol and Gly, respectively [37]. Additionally, this figure shows for muscimol that these receptors mediate a pronounced hyperpolarization and conductance increase. The V_m_ recording also demonstrates that low millimolar theophylline blocks the GABA_A_R [58] in the presence of muscimol, leading to both reversal of the hyperpolarization plus conductance increase and recovery of rhythm [37]. 

Moreover, we reported that opioids depress network rhythm [36]. Specifically, the μR agonist DAMGO abolishes the LFP and longer-lasting crescendo-like multipeak events occur early during recovery (Figure 7) similar to recovery from ɑ_2_R-mediated inhibition (Figure 8). We studied the latter effects in more detail as dealt with in the next sections.

### 2.4. µR- and ɑ_2_(Auto)R-Mediated LFP Pattern Transformations

Our finding that the LFP burst pattern transforms early during recovery from micromolar DAMGO [36] (Figure 7) indicates that such activity is due to an effect of a time period of lower doses occurring in the LC during washout. Indeed, 50–150 nM of either DAMGO or the natural μR agonist morphine typically slow LFP rhythm and evoke the often crescendo-like multipeak LFP burst pattern throughout application periods of several minutes [37] (Figure 7). As one explanation, μR activation may increase the delay between the phase-locked discharge of single spikes. In line with this possibility, LFP amplitude is mostly reduced by DAMGO. As a different or additional mechanism, burster neurons (Figure 4) might discharge spike bursts during this phase. Bursting can principally occur during recovery of the neonatal LC from opioids as exemplified in Figure 7. Here, DAMGO-evoked hyperpolarization was reversed by theophylline, which also antagonizes the response to GABA_A_R or GlyR activation [37] (Figure 6). For muscimol or glycine, recovery of rhythm during washout (or reactivation by theophylline) does not show a phase of multipeak bursting, in contrast to the effect of opioids, despite similar hyperpolarizations. Systemic administration of the μR agonist remifentanil for anesthesia can evoke similarly slow persistent bursting in the LC of rats in vivo [60]. Moreover, intracerebroventricular morphine injection in adult rats evokes even slower neuronal bursts that are reversed by kynurenic acid injection into the LC, which indicates that iGluRs are necessary for this phenomenon [59] (Figure 8). ɑ_2_R activation can also induce spike bursts, an effect observed when interstitial NA levels in the LC of juvenile/adult rat slices were increased by blocking its reuptake with cocaine after spontaneous release. Consequently, cocaine slowed and augmented the amplitude of STOs and transformed single spike discharge into slower discharge of multiple spikes (Figure 8) and these effects were reversed by ɑ_2_R blockade [31]. In agreement, our findings show that LFP bursts are slowed and prolonged during application of 100–250 nM clonidine. Higher clonidine doses block LFP rhythm and (crescendo-like) multipeak bursts occur transiently again during washout (Figure 8). Accordingly, the effects of μR and ɑ_2_R activation on LC rhythm are very similar in the isolated neonatal rat LC. 

### 2.5. Ca_i_ Changes in LC Neurons and Astrocytes

Our results above are based on LFP recording combined with monitoring of either cell-attached single neuron spiking or V_m_. The following findings additionally involved population Ca_i_ imaging of LC neurons and astrocytes. For this, cells are stained with the membrane-permeant chemical Ca^2+^ dye Fluo-4 via pressure injection from a broken patch electrode into the LC center at a depth of ~50 μm into the slice [61,62,63].

In our study on the functional TARP-AMPAR complex [35], the partial agonist CNQX evokes a modest and uniform Ca_i_ rise in neurons with no effect on astrocytes. However, spontaneous Ca_i_ rises occur randomly in a subpopulation of ~20% of astrocytes. This is shown in Figure 9 which also exemplifies that the CNQX-evoked neuronal Ca_i_ rise persists in TTX which lowers Ca_i_ baseline. 

In contrast, we found that 25 μM NA causes a steady decrease of neuronal Ca_i_ baseline which recovers within <3 min upon start of washout. An opposing effect is seen in astrocytes during NA where Ca_i_ increases in a novel type of ‘concentric’ Ca^2+^ wave. Essentially, Ca_i_ increases firstly in astrocytes located >100 μm distant to the LC, then in astrocytes located closer to the LC soma area, and finally in astrocytes within the nucleus [20] (Figure 9) (see Appendix A). The NA-evoked concentric Ca^2+^ wave is blocked by the specific ɑ_1_R antagonist prazosin whereas the neuronal fall of Ca_i_ is blocked by the specific ɑ_2_R antagonist yohimbine. Both neurons and astrocytes seem to possess metabotropic glutamate receptors and purinergic receptors because the respective agonists t-ACPD and adenosine triphosphate (ATP) cause a notable Ca_i_ rise in astrocytes (mostly also in a concentric wave for ATP, but not t-ACPD) whereas neuronal responses are smaller [20] (Figure 9). The neuronal Ca_i_ changes are apparently modest as application of 0.5 or 1 mM glutamate at the end of an experiment causes a >5-fold larger increase in Fluo-4 fluorescence. As a further observation, the number of astrocytes seems to be smaller within the nucleus compared to surrounding slice areas [35] (Figure 3).

With population imaging, covering the LC and portions of surrounding tissue, we did not detect LFP-related neuronal Ca_i_ increases in control, contrary to spike-related (dendritic) Ca_i_ rises seen in adult mouse slices using fast line-scanning imaging [46] (Figure 10). However, we detect rhythmic Ca_i_ rises in solution with 7 or 9 mM when the LFP transforms to slower pronounced multipeak bursts as we have reported previously [36] (Figure 10). 

For our above findings, we used a bath-application approach to apply neuromodulator agonists and antagonists. While this partly mimics global autocrine modulation of the LC network, it is not clear how cells respond to activation of afferent inputs. It was mentioned above that single electrical stimuli of areas surrounding the LC evoke an EPSP-IPSP sequence [38]. We were unable to detect changes in neuronal Ca_i_ in response to a single stimulus, contrary to occurrence of robust signals in response to repetitive stimulation involving >10 single pulses at a rate of 20–100 Hz [20]. Such ‘tetanic’ stimulation causes only a rise of Ca_i_ in ~60% of neurons whereas in the remaining cells the initial Ca_i_ increase is followed by an ‘undershoot’ below its baseline (Figure 10). A major portion of the stimulus-evoked Ca_i_ rise is attenuated by CNQX and the remaining portion by prazosin [20] (Figure 10). 

The last aspect of the present study deals with a novel neuron-astrocyte interaction that has been detected in cultured LC slices from newborn rats [64]. Using a combined optogenetic/Ca_i_ imaging and electrophysiological approach, the authors demonstrated a new role of the metabolite L-lactate in neuron-glia communication. Specifically, depolarization of LC astrocytes causes them to release L-lactate, which then diffuses to neighboring neurons to evoke a depolarization that initiates their spiking via a yet unknown receptor (Figure 1). We also find that in acutely isolated newborn rat slices, L-lactate acts excitatory to accelerate LFP rhythm likely in response to ɑ_1_R-activated vesicular Ca_i_ rise in astrocytes. 

## 3. Discussion

By combining electrophysiological recording with population Ca_i_ imaging, we present here evidence for novel neonatal LC properties. We demonstrate that the newborn rat LC forms a neural network that is capable of generating a LFP. In the following sections we summarize the properties of this LFP and refer them, on the one hand, to those detected in single neurons based mainly on in vitro studies in newborn, young, and adult rodents. On the other hand, we compare neonatal LC network properties to those in the adult mammalian LC showing a modular organization.

### 3.1. Intrinsic Neonatal LC Properties

LFP recording in vivo and in brain slices was, and still is, instrumental for unraveling neural network properties in a variety of brain regions, with the largest number of studies on the cortex, including hippocampus [40,41,42,63,65,66,67,68]. For the LC, no LFP has been reported in slices until our first study did so [22]. In that study and our follow-up work [36], we showed that recording the LFP depends on using ~50 µm large suction electrodes instead of often used fine-tipped (patch) or metal electrodes. Suction electrodes are typically applied for in vitro nerve root recording, mostly in models of the neonatal rodent locomotor or respiratory networks [69,70,71]. The group of Ramirez also possibly found out that they can be used to monitor the LFP generated by rhythmogenic inspiratory center neurons from the surface of the corresponding ventrolateral area in ‘breathing slices’ [71,72].

LFP shape and amplitude do not appear to differ between recording sites within the LC. This indicates that there are no local neuron modules that may discharge under control conditions in a fashion basically different from that of the main LC network ensemble as seen in the inferior olive of slices from young and adult rats [73] and LC [15,17]. Our finding of a blocking effect of TTX on both the LFP and STOs is similar to observations in the en bloc model [21] and adult mouse slices [46], whereas TTX had mostly no effect on STOs in other LC slice studies [31,32,34].

The LC-LFP comprises spike discharge of 3–10 LC neurons with no synaptic component, contrary to that in the hippocampus [40,65,74]. The ~300 ms-lasting bell-shaped signal that the LFP represents is due to the fact that typically single spike discharge in each neuron shows a jitter and is phase-locked to a particular time period within the LFP (Figure 3). Both phenomena are perhaps at least partly due to the consequence of a different ‘resting’ V_m_ in individual neurons in combination with random occurrence of spontaneous subthreshold PSPs delaying or shortening the time point when V_m_ reaches spike threshold (Figure 3), as we have hypothesized previously [36]. Our findings show that spiking in the LC neuron network is not synchronous. This prompted us to conclude that electrical coupling via gap junctions is not strong enough to cause full synchrony of the network as has been discussed previously [13,27,34,51,52]. It seems that the extent of synchrony can be increased or decreased further by neuromodulators as discussed below.

While the rate of phase-locked single spike discharge in newborn rat slices is similar in a given slice with an average of ~1 Hz, intrinsic neuronal membrane properties differ. In newborn rat brainstem–spinal cord preparations, type-1 and type-2 LC neurons differ in STO amplitude and regularity [21] while we found that pacemaker, intrinsic burster, and irregular LC neurons can be discriminated by current-evoked depolarization (Figure 4). According to our knowledge, adult LC neurons show exclusively pacemaker discharge in control. It is important to note that morphine application changes spiking from tonic to bursting in adult rat LC neurons in vivo [59] (Figure 8), but it is not clear whether this spike pattern transformation is caused by depolarization-related intrinsic bursting. It can also be due to changed synaptic input as kynurenic acid reverses this spike pattern transformation [59]. In that regard, during opioid withdrawal, complex interactions occur between µR, OXR1, and iGluR within the orexinergic-opioidergic system, particularly for connections between the nucleus paragigantocellularis and the LC [75,76,77,78] (Figure 1). Intrinsic bursting in neonatal rat LC neurons is less pronounced than in other neuron types, such as in the inspiratory center [71] or cortex [79]. This might partly be due to dialysis of cellular constituents such as cAMP during whole-cell recording, as shown for LC neurons in the newborn rat en bloc model [21] and in adult rat slices [80]. 

Ion conductances that can be involved in this endogenous bursting are often persistent voltage-gated Na^+^ channels, which seem to be functional in the neonatal LC [22] (Figure 3), different types of voltage-gated Ca^2+^ channels, and Ca^2+^-activated K^+^ channels. In adult rat slices, spontaneous (single) spikes or stimulus-evoked spike trains cause activation of voltage-gated Ca^2+^ channels followed by an after-hyperpolarization mainly mediated by Ca^2+^-activated K^+^ channels [81]. In adult mouse slices, the role of voltage-gated Ca^2+^ channel and Ca^2+^-activated K^+^ channel (subtypes) in LC neuron pacemaker behavior has been studied in detail [47,48]. It is not clear yet whether the different neonatal neuron types are located in a particular ‘module’ area with the neonatal LC and/or project to a specific (group of) target brain areas like in the adult LC [3,5,15,17].

### 3.2. Independence of LC Network Rhythm on Anion Channel-Mediated Inhibition and iGluR

Regarding ‘classical’ inhibition, blockade of GABA_A_R or GlyR causes a rhythm change to seizure-like bursting in the adult and newborn cortex and hippocampus [63,65,74]. Similarly, alternating spinal locomotor activity changes to seizure-like synchronous bursting during blockade of these receptors [70,82]. In contrast, the primary process of inspiratory rhythm generation in the brainstem does not depend on such inhibition [83,84,85,86]. Regarding glutamatergic synaptic transmission, blockade of iGluRs abolishes rhythm in the inspiratory center [69,71], locomotor central pattern generator [87], and other brain circuits [43,54,63,66,73].

Previous in vitro studies using adult rat slices have established that single neuron spiking in the LC persists during blockade of inhibition via GABA_A_R and GlyR [88] and of excitation via iGluR [89,90]. These and related studies also showed that GABA_A_R and GlyR activation abolishes single LC neuron spiking whereas iGluR agonists increase their spike rate [89,90,91]. Our LFP and V_m_ recordings revealed that neonatal rat LC neurons already possess functional GABA_A_R, GlyR, and iGluR, while blocking these receptors does not affect network rhythm. 

It can be concluded that neither GABAergic interneurons nor potential tonic release of GABA or glycine from afferent axon terminals within the isolated newborn rat LC are necessary for rhythm. The same can be concluded for eventual spontaneous release of glutamate within the LC in such slices.

### 3.3. iGluR-Mediated LFP Pattern Transformation

We found that the competitive AMPAR and KAR blocker CNQX accelerates the rhythm whereas the specific competitive AMPAR blocker GYKI or the non-specific, non-competitive iGluR blocker kynurenic acid have no effect [35]. The CNQX effect is due to the agent acting as a partial agonist on LC neuron AMPARs that are coupled to TARP. As many spontaneously active neural networks depend on active AMPAR-containing synapses, it follows that the yet inhibitory CNQX effect on transmission in these networks abolishes the rhythm despite a potential stimulatory agonistic action on V_m_ of individual neurons [43,54,66,69,71,73,83,87]. As the neonatal LC rhythm does not depend on iGluR, we showed for the first time that activation of the TARP–AMPAR complex has a stimulatory effect on a spontaneously active neural network. 

In some slices from our study on the TARP–AMPAR complex [35] we noticed that the amplitude of LFP rhythm fluctuates during CNQX-mediated stimulation. Similarly, AMPA, KA, and NMDA merge separate bursts into faster sinusoidally-shaped oscillations showing spindle-like amplitude variations. As one possible explanation, this transformed LFP pattern might represent summation of enhanced STOs in LC neurons that are strongly electrotonically coupled in newborn rats [26,32,34,44]. As examples for this mechanism in neonatal rats, KA or the cholinergic agonist carbachol induce STOs that summate to spindle-shaped γ-type LFP oscillations in the CA3 hippocampal area. These oscillations are, however, notably faster than those in the LC [40,66,92]. Similarly, STOs are involved in sleep-related γ spindles in the adult brain [93,94]. Moreover, most inferior olive neurons, which share various properties with the neonatal LC, such as strong coupling via gap junction, show rhythmic variations in frequency and amplitude of STOs [73]. As STO variations were blocked by CNQX in that juvenile rat brain slice study, the authors concluded that tonic depolarizing glutamatergic input is involved in this phenomenon. Indeed, we found that during AMPA or KA LC neurons show a modest depolarization and ‘tonic’ increase in spike rate with no rhythmic amplitude changes of either spikes or underlying STOs. Possibly, the spindle-shaped amplitude LFP fluctuations are due to a rhythmic change in the extent of phase-lock of individual neuron discharge regarding the network output [66] and this increase in phase-lock towards higher synchrony can be due to an enhanced coupling of gap junctions [95]. In that regard, NMDA stabilizes synchronized LFP oscillations in gap junction-coupled inferior olive neurons of rat slices. This process involves Ca^2+^/calmodulin-dependent protein-kinase-I activation which enhances weak coupling of non-neighboring neurons [96]. In contrast, in the adult rat LC, neuromodulators (including glutamate) presumably do not directly counteract the postnatal decrease of gap junction coupling and, instead, neuromodulator-evoked spike slowing itself reverses this decrease [34]. We found that NMDA-evoked acceleration of rhythm is accompanied by increased synchrony which we could not yet confirm for the network oscillations due to AMPA and KA. In each case, though, all three agents increased the regularity of LFP rhythm.

In neonatal LC neurons, NMDA causes crescendo-shaped rhythmic sustained depolarizations and concomitant spike discharge followed by hyperpolarizations that are closely related timewise to the oscillation train LFP pattern (Figure 5). Detailed pharmacological analysis is required to analyze cellular mechanisms underlying NMDA-evoked bursting that can include modulation of Ca^2+^-dependent STOs and a variety of other processes [97,98,99,100]. While many questions remain regarding the iGluR agonist-evoked LFP pattern transformations, our findings support the hypothesis based on in vivo findings of morphine-evoked bursting in the LC of adult rats that iGluR within the LC is needed for the spike pattern transformation of its neurons by opioids [59] (Figure 8).

### 3.4. LFP Pattern Transformations by μ-Opioid and α_2_ Receptors

The very similar complex inhibitory effects of μR and α_2_R activation on the newborn rat LFP [36] (Figure 7 and Figure 8) are not surprising as they act on the same G_i/o_-mediated cellular signaling pathway coupled to GIRKs [101,102]. For slice studies on the LC, mostly high doses were used that result in blockade of neuronal spike discharge due to pronounced GIRK channel-mediated hyperpolarization. We found that low morphine, DAMGO, or clonidine doses slow LFP rhythm and evoke a (crescendo-like) multipeak burst pattern. As one explanation, low doses of these μR and α_2_R agonists may have a modest hyperpolarizing action on LC neurons. Due to the differences in resting V_m_ and subthreshold EPSPs and IPSPs, this might increase the differences in time points at which each neuron reaches spike threshold. Additionally, the possibility of presynaptic effects cannot be excluded. However, as noted above, LFP rhythm does not depend on classical synaptic processes and may rather rely on intrinsic STOs and gap junction coupling. While the observed LFP pattern transformation occurred only in a narrow nanomolar dose range of about one order of magnitude, it is not clear which concentrations are reached in the LC of adult rats in vivo when μR agonists are applied (systemically) to cause slow LC neuron bursting [60,74]. Additionally, in the latter studies it is not clear whether the agonists act directly on the LC or rather on afferent circuits which then induce synaptically-mediated bursting (see above). It is also unknown in which way iGluRs act to support such bursting [59] (Figure 8). These considerations also raise the question of which opioid and clonidine doses occur in the LC during their recreational and therapeutic effects. In that regard, under very similar in vitro conditions, ~100 nM DAMGO or fentanyl block rhythm in the isolated inspiratory center [58,71]. 

### 3.5. Ca_i_ Responses in the Neonatal LC

Our immunohistochemistry [35] and Ca_i_ imaging results in newborn rat slices [61] showed firstly that LC neuron somata were densely packed whereas somata of intermingled astrocytes were notably fewer and smaller (Figure 2). Despite a smaller number of somata, astrocyte processes might spread throughout the LC and may contain the receptors needed for their communication with neurons. One novel type of communication is via lactate, which seems to be released upon α_1_R activation from astrocytes and diffuses to neighboring neurons to excite them via a novel receptor [10,64] (Figure 1). While the latter findings were obtained in cultured LC slices, we made similar observations regarding lactate in the acutely isolated newborn rat slices. We also found that the presumptive astrocytes respond to various neuromodulators in a fashion distinct to neurons. Specifically, NA decreased Ca_i_ baseline in neurons while inducing a concentric Ca_i_ wave starting in the dense network of astrocytes surrounding the LC (Figure 9). DAMGO also decreased baseline Ca_i_ in neurons while it did not affect astrocytes. The astrocytic Ca_i_ increases during NA are likely due to metabotropic Ca^2+^ release from endoplasmic reticulum stores and this has been shown to occur in several brain regions in wave-like fashion based on gap junction coupling [9,61]. In contrast, the DAMGO- and NA-evoked neuronal Ca_i_ decreases are, at least to a major extent, likely due to decreased Ca^2+^ influx associated with the hyperpolarization and concomitant spike blockade, resulting in inactivation of spike-related and persistently open (‘tonic’) voltage-gated Ca^2+^ channels. Our observation that ATP and t-ACPD also cause a notable (wave-like) Ca_i_ rise in astrocytes vs. a modest response in neurons indicates that neurons do not have a major amount of Ca^2+^ stored in the endoplasmic reticulum and/or may not have purinergic or metabotropic glutamate receptors for the latter agonists, respectively.

Ca_i_ baseline in neonatal LC neurons also increases modestly with similar magnitude and kinetics in all analyzed neonatal LC neurons during CNQX-evoked acceleration of rhythm [35]. These Ca_i_ rises are likely mainly due to enhanced activation of tonic voltage-gated Ca^2+^ channels, possibly in concert with a minor influx through the potentially Ca^2+^-permeable AMPAR [53,103]. We proposed that most LC neurons express TARP–AMPAR complexes or that they are at least in some neurons secondary to gap junction coupling, causing a similar depolarization of neighboring cells lacking a functional TARP–AMPAR complex [35]. 

The fact that no rhythmic neuronal Ca_i_ rises were seen during single spikes or crescendo-like multipeak bursts during recovery from high DAMGO or NA is likely caused by the limited time resolution of our scanning population imaging approach at ~1 frame/s. Accordingly, rhythmic single spike-related (dendritic) Ca_i_ rises were recorded in LC neurons from adult mouse slices using fast line-scan imaging [46] (Figure 10). With our approach, rhythmic neuronal Ca_i_ rises were only seen during sustained LFP bursting in 7 or 9 mM K^+^ solution (Figure 10). Moreover, robust neuronal Ca_i_ rises occurred in response to tetanic stimulation of the pericoerulear areas surrounding the LC soma region. While the blocking effects of CNQX and prazosin (Figure 10) indicate that the Ca_i_ rises in most stimulation sites involve AMPAR/KAR and α_1_R activation, respectively, it is possible that stimulation of other (more remote) sites reveals other synaptic inputs to the LC network. As an important new result, stimulation evoked either only a Ca_i_ rise or a rise followed by an undershoot below baseline. The undershoot likely involves α_2_R activation based on findings from V_m_ recording in adult rat slices showing that single electrical stimuli cause an iGluR mediated EPSP followed by a prolonged α_2_R-mediated afterhyperpolarization [38]. The finding of two different neuronal response types to tetanic stimulation in neonates is further indication of non-uniform neuronal properties similar to those of the adult LC. 

In summary, our Ca_i_ imaging findings are a first important step to elucidate receptors and signaling pathways in the neonatal LC. In the next section, we refer to how our in vitro approaches can be complemented by other approaches in future studies and how the findings relate to the current understanding of the modular organization of the adult LC.

### 3.6. Conclusions and Perspective

The previous and current findings on the isolated newborn rat LC indicate clearly that this neural network is already very complex at birth and also possibly has a modular organization such as that in adults. Further novel neonatal LC properties will certainly be detected with the in vitro techniques used so far, specifically bath-application of drugs, LFP recording, cell-attached plus whole-cell spike and PSP analysis, and population Ca_i_ imaging. However, other approaches discussed in the following will likely lead to other novel findings. For several years, in vivo studies in the adult rodent LC have used powerful approaches including optogenetic and chemogenetic stimulation in genetically-engineered animals, sometimes combined with subsequent in vitro analyses, e.g., electrophysiological recording and pharmacology in slices or immunohistochemistry [17,64,104,105,106]. However, genetic engineering in newborns is only a newly emerging field as the manipulation must occur in utero [107]. Furthermore, acute approaches such as drug injection or electrophysiological recording, e.g., with multi-site electrodes [19,108,109] are challenging in the very small neonatal LC. Accordingly, we focus here on future in vitro techniques. 

Regarding pharmacology, bath-application of drugs partly mimics autocrine LC neuromodulation. This holds particularly true for NA (ant)agonists because strong excitatory afferent inputs to (a subpopulation of) LC neurons increases the spike rate not only of these cells, but likely also neighboring neurons due to gap junction coupling. The increased spiking would then cause a global NA release within the LC that then has feedback effects on both neurons (via auto α_1_R and α_2_R) and astrocytes (via α_1_R). Modest effects of, e.g., iGluR, GABA_A_R, or μR activation might be more localized if neonatal LC neurons have a modular organization regarding expression of such receptors. To analyze this, drugs can be focally injected into LC subareas during electrophysiological LFP and/or cellular recording. In that regard, it can be tested whether findings from drug injection into the LC soma region differs from those upon focal application to the pericoerulear region where autocrine synaptic integration is prominent [44,110]. Regarding recording, multi-electrode arrays [111] would be suitable to detect, e.g., whether some LC neurons respond with bursting to focal drug application. Moreover, whole-cell recording of V_m_ changes (and underlying ion currents in voltage-clamp) can be combined with imaging of cellular factors such as Ca_i_ or cAMP under the influence of drugs. After whole-cell recording, the cytoplasm can be harvested for identifying via PCR analysis neuron type-specific ion channels or receptors [112]. Specific afferent inputs to (subpopulations of) LC neurons can be identified via electrical stimulation in LC slices sectioned at an angle that would preserve axon tracts from remote brain areas. In that regard, focal LC stimulation would possibly reveal that modules within the LC respond with LFP and possibly also V_m_ pattern transformation to iGluR, μR, OX1R, or α_2_R activation. One promising rat slice model in that respect retains functional connectivity from neurons of the nucleus paragigantocellularis to those in the LC [77]. It would also be important to use slices in which LC output tracts are preserved to identify whether particular axons propagate spike bursts during LFP pattern transformations evoked by the latter receptors. Regarding the LFP pattern transformation, data from simultaneous LFP or multi-electrode array recording combined with monitoring single neuron V_m_ changes can be used for modeling, e.g., of spindle-shaped AMPAR- and KAR-mediated LFP oscillations whose shape resembles γ-oscillations elicited by bath-applied KA in cortical slices which are thought to provide a temporal structure for information processing in the brain [66]. As examples for topics to be studied with such approaches, one could analyze how iGluR, μR, and OX1R eventually cooperate to transform LC neuron LFP and possibly also single neuron discharge patterns like in adult rats in vivo [59]. Regarding modeling, approaches like those used in the inferior olive may be applied. Specifically, it has been shown that two principal characteristics of its neurons, i.e., STO and electrical gap junctions, make this system a powerful encoder and generator of spatiotemporal patterns with different but coordinated oscillatory rhythms [113]. 

## Figures and Tables

**Figure 1 brainsci-12-00437-f001:**
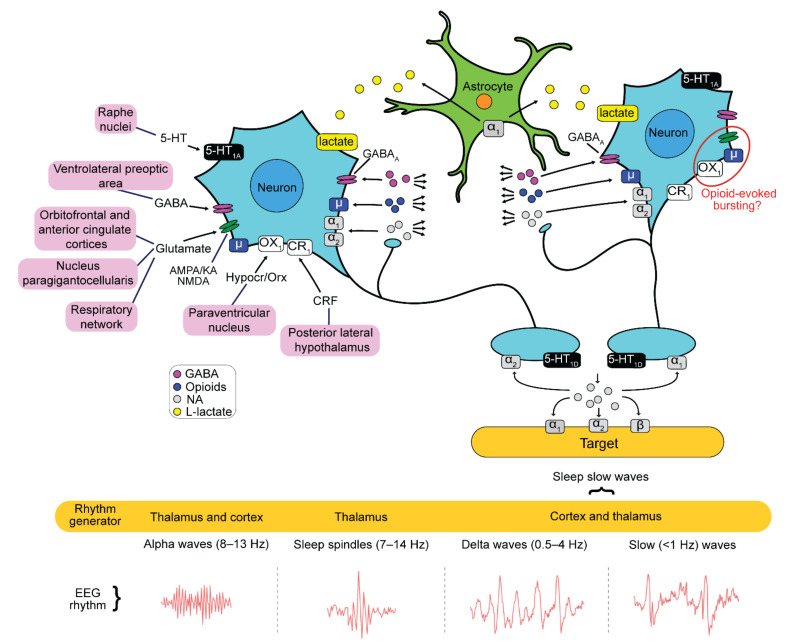
Adult locus coeruleus (LC) connectivity. The LC in adult mammals receives synaptic input from diverse remote brain circuits exemplified in the upper left section. The input involves a variety of neurotransmitters acting on different receptor (R) subtypes, e.g., such as glutamate on ionotropic R-subtypes (iGluR) activated by either α-amino-3-hydroxy-5-methyl-4-isoxazolepropionic acid (AMPA), kainate (KA), and N-methyl-D-aspartate (NMDA), γ-aminobutyric acid (GABA) on GABA_A_R, serotonin (5-HT) on 5-HT_1A_R or 5-HT_1D_R, peptides such as endogenous opioids on μ-receptors (μR), hypocretin (Hypocr), orexin (Orx) acting, e.g., on OX1R and corticotropin-releasing factor (CRF). As shown in the middle right and lower section, LC neurons release in remote brain areas in most parts of the neuraxis their main neurotransmitter noradrenaline (NA) which acts on either α_1_, α_2_, or β R-subtypes, and possibly a LC neuron subtype-specific co-transmitter such as the peptide galanin or neuropeptide-Y. By activity-related NA and co-transmitter release, the LC controls (spontaneous) activities of brain circuits, e.g., in the thalamus or cortex, to modulate electroencephalogram (EEG) patterns or behaviors such as sleep. At the same time, some of the neurotransmitters act in feedback fashion on presynaptic synapses on the same LC neuron. Importantly, some LC neuron collaterals terminate within the LC to release NA and their co-transmitter for acting on neighboring neurons or astrocytes shown in the upper section. Astrocytes possess, for example, α_1_R whose activation might cause release of the metabolite L-lactate that then stimulates LC neurons via a novel R type. Neuron–neuron and neuron–astrocyte interactions serve for autocrine control within the LC. Note that the red circle in the LC neuron in the upper right part indicates one example for an interaction between postsynaptic Rs. Specifically, μR might interact with iGluR and OX1R during opioid actions. All these functions, and other properties described in the main text, show that the adult LC has a complex connectivity and modular organization for enabling its diverse control functions. The schema is partly based on that from [7]. Schema of EEG recordings adapted with permission from [11].

**Figure 2 brainsci-12-00437-f002:**
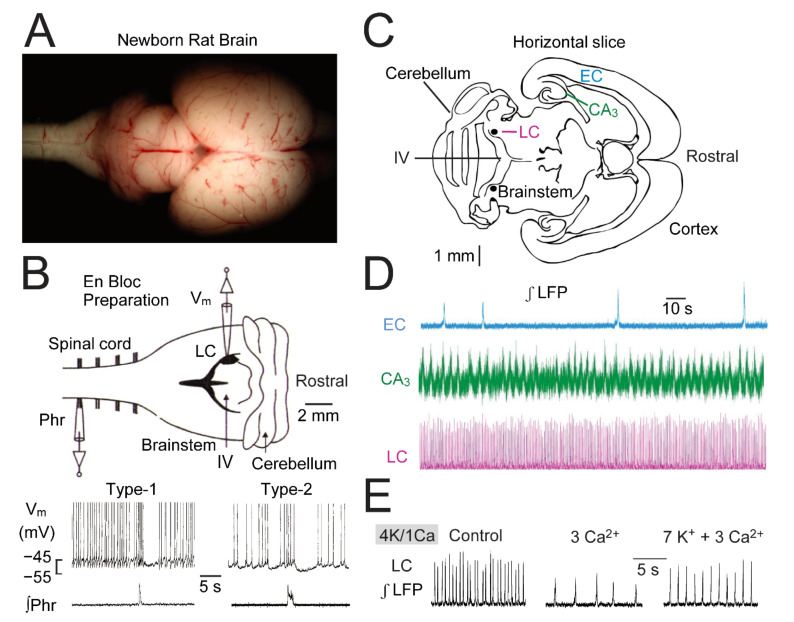
Spontaneous rhythms in LC, cortex and hippocampus of horizontal newborn rat brain slices. (**A**) The image in the upper left section shows a dorsal view on the isolated brain and parts of the cervical spinal cord. (**B**) Displays in upper part a schema for simultaneous suction electrode recording of integrated inspiratory-related phrenic nerve (Phr) activity from a ventral spinal rootlet and whole-cell membrane potential (V_m_) recording from neurons in the LC indicated as a black oval located close to the 4th ventricle (IV). The lower part illustrates that spontaneous inspiratory phrenic bursts evoke in both neurons an initial acceleration of spontaneous LC neuron action potential (‘spike’) discharge followed by a hyperpolarization and concomitant spike blockade lasting ~5 s. In the ‘type-1′ neuron on the left, subthreshold oscillations (STOs) of V_m_ and concomitant spike discharge are very regular, contrary to more irregular STOs and spiking in the ‘type-2’ neuron in the right (**C**) Schema of a 400 µm thick horizontal slice at a section level that enables simultaneous recording of spontaneous local field potential (LFP) rhythms in entorhinal cortex (EC), CA_3_ hippocampal area (CA_3_), and LC brainstem area. (**D**) Simultaneous suction electrode recordings from the surface of a slice kept in superfusate containing (among other components) 4 mM K^+^ and 1 mM Ca^2+^ (‘4K/1Ca’). LFPs were differentially amplified (×10 k) and band-pass filtered (0.3–3 kHz) and integrated at a time constant of 20 ms using a ‘moving averager’. Note that rhythms have a rate of 1–4 bursts/min in EC, 10–40 bursts/min in CA_3_ area and 0.5–3 Hz in LC. (**E**) An increase of superfusate Ca^2+^ from physiological 1 to 3 mM, almost abolishes the LC LFP. Use of elevated superfusate Ca^2+^ might partly explain why LFPs have not been recorded previously in LC. Note that our group uses typically a 3K/1.2Ca superfusate. (**A**) from [20]; (**B**) adapted with permission from [21]; (**C**–**E**) adapted with permission from [22].

**Figure 3 brainsci-12-00437-f003:**
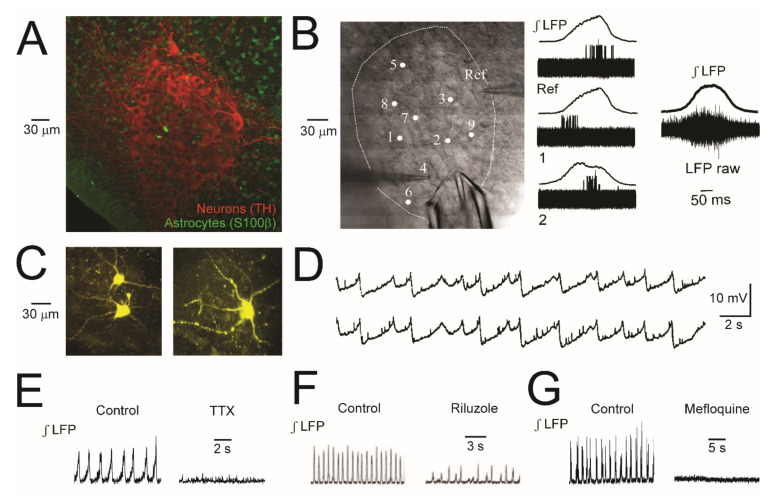
Cell morphology, ‘phase-locked’ spiking and related basic neuronal properties in newborn rat LC slices. (**A**) Image shows in a chemically fixed slice with double immuno-histochemical staining that the LC is comprised of ~90% densely packed tyroxine-hydroxylase (TH) -positive neurons with a mostly >20 µm soma diameter while smaller cells are glial cells, mostly S100β-positive astrocytes. (**B**) Traces next to image show overlay of 20 cycles of averaged integrated suction electrode-recorded LFP traces monitored simultaneously with single neuron spiking detected with a ‘cell-attached’ patch electrode. Spiking in a reference neuron (Ref) was continuously monitored with consecutive recording in a further 9 neurons (cells 1 and 2 displayed here). Traces on the right show the averaged integrated LFP and overlaid raw LFP signals from all cycles. Such spike tracking revealed that LC neurons discharge (with a ‘jitter’) during a particular phase of the LFP comprising overlaid spiking of 3–10 neurons. (**C**) The morphology of neurons filled with lucifer-yellow via the patch electrode during whole-cell V_m_ recording. (**D**) Synchronous V_m_ STOs in 2 simultaneously recorded LC neurons. The smaller and shorter V_m_ depolarizations may reflect spontaneous postsynaptic potentials (PSPS). (**E**) LFP rhythm is abolished by bath-application of the voltage-gated Na^+^ channel blocker tetrodotoxin (TTX, 50 nM) or the connexin-32 gap junction blocker mefloquine (100 μM). (**A**) adapted with permission from [35]; (**B**,**E**–**G**) adapted with permission from [36]; (**C**) from [37]; (**D**) adapted with permission from [32]; Copyright (1989) Society for Neuroscience.

**Figure 4 brainsci-12-00437-f004:**
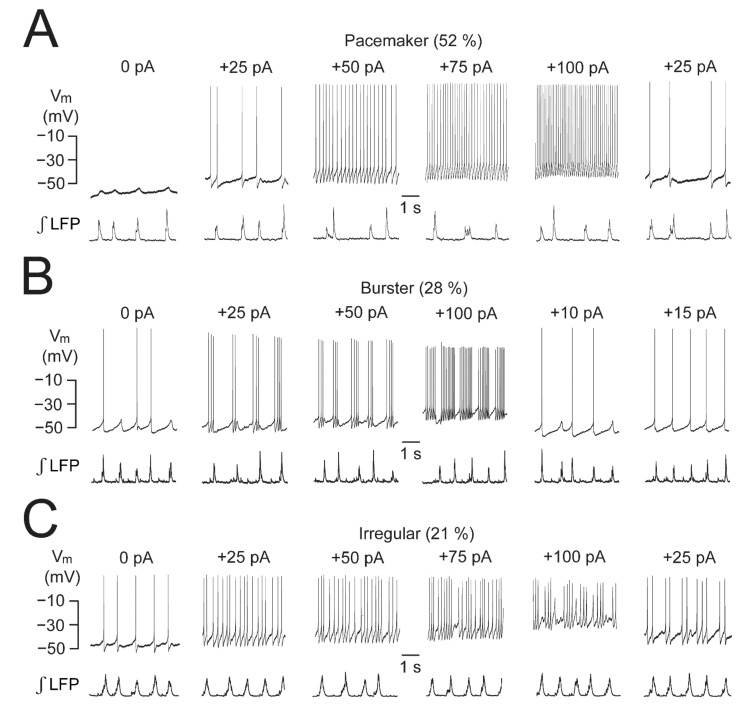
Intrinsic spike patterns in LC neurons of newborn rat slices. (**A**) Depolarization of a whole-cell-recorded ‘pacemaker’-type neuron by current injection through the patch electrode shows a gradual increase in the rate of very regular spiking between 25 and 100 pA. Note that this cell shows only STOs in control due to its quite negative resting V_m_. (**B**) Depolarization of a ‘burster’-type neuron evokes groups of spikes whose number increases with the extent of depolarization. (**C**) Depolarization of an ‘irregular’-type neuron only modestly increases spike rate at 25 to 100 pA. Numbers indicate the percentage of occurrence of these neuron types based on 29 recordings. For our whole-cell V_m_ recordings (see also Figure 5A,C, Figure 6C, Figure 7C), access resistance was compensated during a test pulse at the beginning of a recording and was also checked, and eventually adjusted, later during the measurement. Access resistance typically ranged between 10–50 MΩ and was stable in >95% of neurons even during recordings lasting > 1 h. For determining neuronal input resistance ranging from 120–370 MΩ, hyperpolarizing current pulses (50–100 pA) were injected, mostly at an interval of 10–15 s (see Figure 6C and Figure 7C). From [20].

**Figure 5 brainsci-12-00437-f005:**
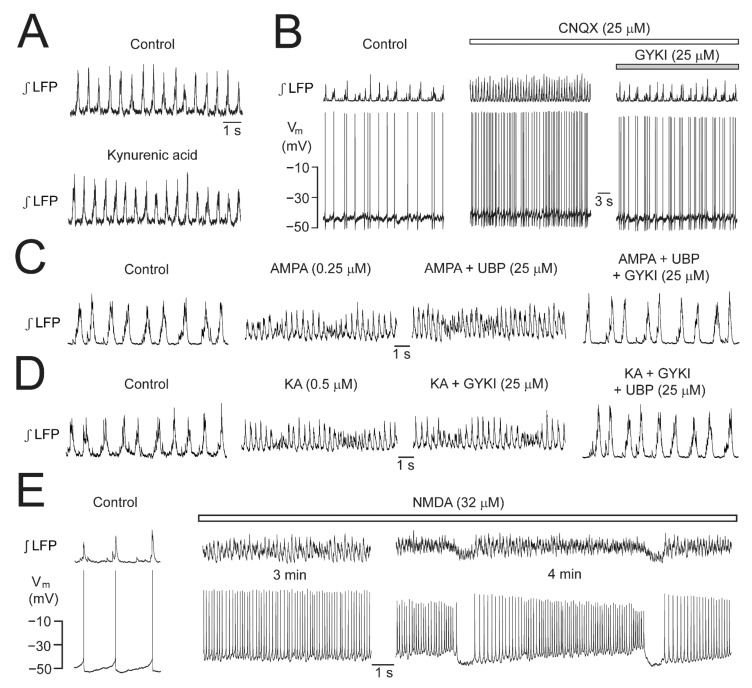
iGluR (ant)agonist effects on LC activities in newborn rat slices. (**A**) Bath-application of 2.5 mM of the broad-spectrum competitive iGluR blocker kynurenic acid for 5 min does not affect LFP. (**B**) Acceleration of LFP rhythm by bath-applied 6-cyano-7-nitroquinoxaline-2,3-dione (CNQX) is accompanied by modest V_m_ depolarization leading to faster cellular spiking. These stimulatory CNQX effects are reversed within 2 min after start of adding 25 µM GYKI to the CNQX-containing solution. (**C**) Bath-applied AMPA evokes fast LFP oscillations with spindle-shaped amplitude fluctuations that persist after adding the KAR antagonist UBP-302, but are blocked by further addition of GYKI. (**D**) Bath-applied KA evokes very similar LFP oscillations that persist after adding GYKI, but are blocked by further addition of UBP. (**E**) Several minutes after start of NMDA application, V_m_ oscillations become interrupted by ~1 s-lasting rhythmic hyperpolarizations causing spike blockade for ~1 s. The resulting LFP oscillation trains start after the inactivity phase with concomitant progressive neuronal depolarization leading to accelerated spiking (right panels). (**A**,**B**) Adapted with permission from [35]; (**C**,**D**) from [49]; (**E**) from [50].

**Figure 6 brainsci-12-00437-f006:**
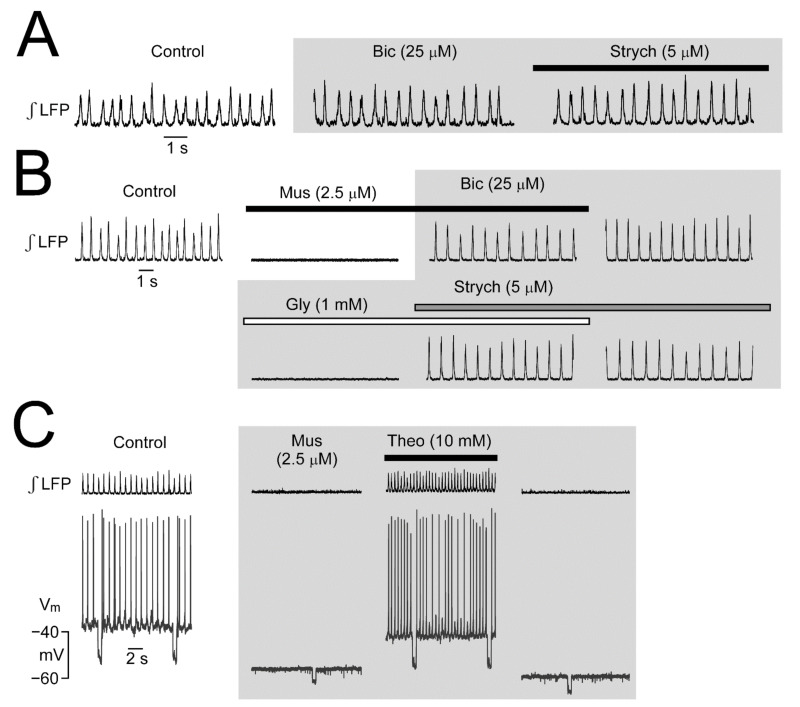
GABA_A_ and glycine (Gly) receptor (ant)agonist effects on LC activities in newborn rat slices. (**A**) Bath-application of the GABA_A_ receptor blocker bicuculline (Bic) has no effect on LFP and even addition of the Gly receptor blocker strychnine (Strych) to this solution does not perturb rhythm. (**B**) The neonatal rat LC has functional GABA_A_ and Gly receptors as the GABA_A_ receptor agonist muscimol (Mus) abolishes rhythm which is restored by adding Bic to the Mus-containing solution. Subsequent bath-application of Gly in Bic-containing solution also abolishes rhythm that is then reactivated by Strych. (**C**) Blockade of LFP rhythm by bath-application of Mus is accompanied by a LC neuron hyperpolarization and a decrease of input resistance measured by repetitive injection of hyperpolarizing current pulses at 10 s interval. The effects are countered by addition of theophylline (Theo) to the Mus-containing superfusate. From [37].

**Figure 7 brainsci-12-00437-f007:**
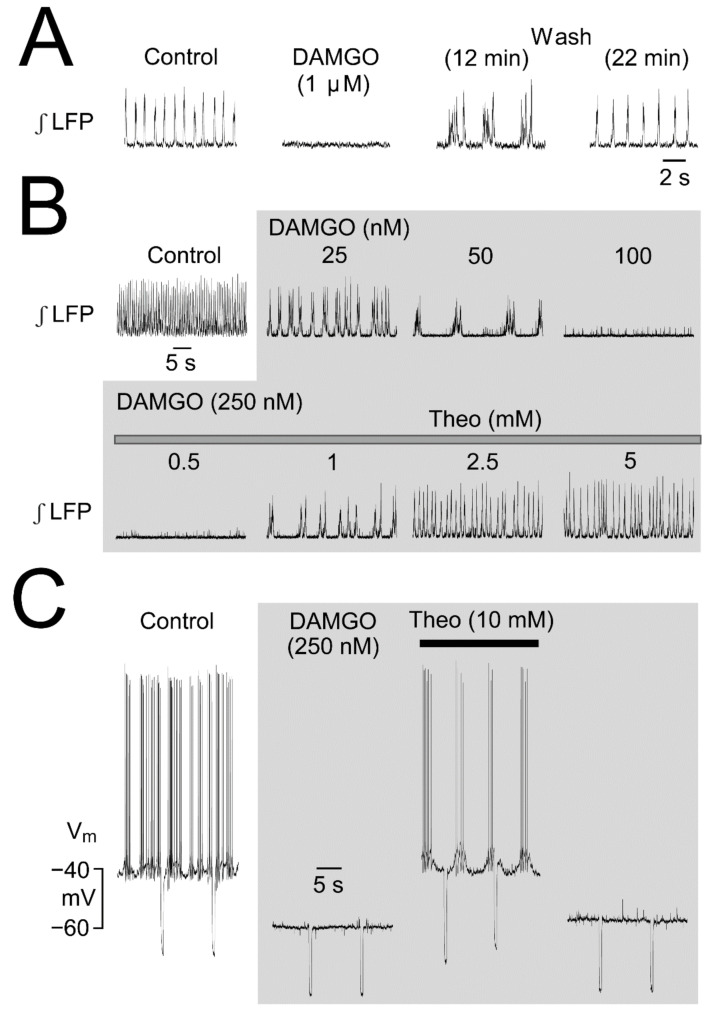
Depressing opioid effects and stimulatory theophylline action on LC activities in newborn rat slices. (**A**) Bath-application of 1 µM of the µR agonist [D-Ala^2^,N-Me-Phe^4^,Gly^5^-ol]-enkephalin (DAMGO) abolishes LFP rhythm. Recovery to the normal pattern of rhythm 22 min after start of DAMGO washout is preceded by a period of occurrence of slower multipeak bursts. (**B**) DAMGO hyperpolarizes V_m_ and abolishes intracellular spiking whereas bath-application of Theo reverses the hyperpolarization and induces rhythmic bursting. (**C**) Bath-application of increasing DAMGO doses transforms LFP pattern into multipeak bursts (at 25–50 nM) while rhythm is abolished at 250 nM. Rhythm recovers upon addition of 1 mM Theo to multipeak bursting whereas 5 mM Theo restores a normal pattern. Input resistance was measured by repetitive injection of hyperpolarizing current pulses at 10 s interval (**A**) adapted with permission from [36]; (**B**,**C**) from [37].

**Figure 8 brainsci-12-00437-f008:**
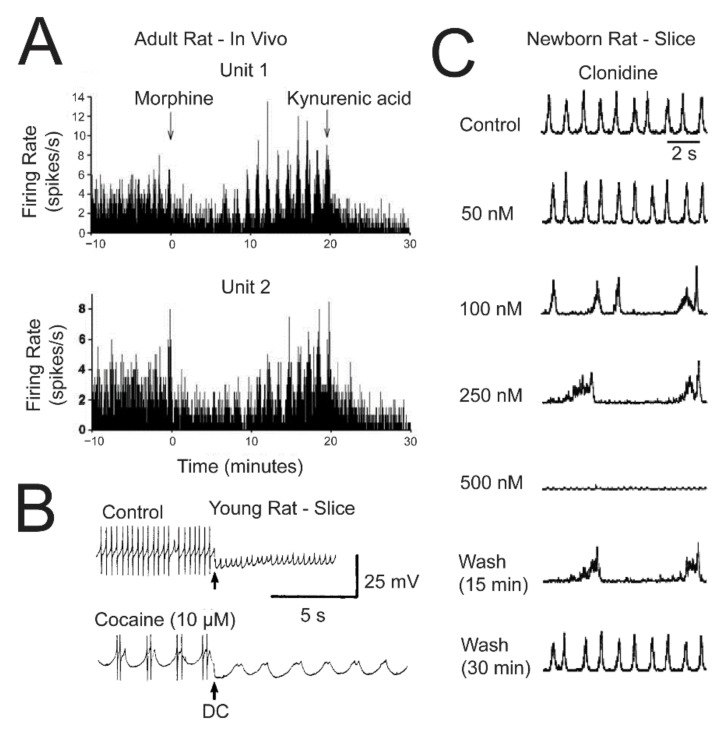
μR- and α_2_R-evoked LC discharge pattern transformations. (**A**) In an adult rat in vivo intracerebroventricular application of the µ-opioid agonist morphine transforms asynchronous low rate spiking of 2 extracellularly recorded LC neurons into slow, synchronous spike bursts. Subsequent kynurenic acid injection reverses this effect. (**A**) In a slice from a young rat, cocaine enhances the amplitude and prolongs the duration of subthreshold oscillation leading to a change in the discharge from 1 to several spikes per event. (**A**) In a newborn rat slice increases in the dose of bath-application of the a_2_R agonist clonidine firstly slow LFP rhythm, then induce crescendo-like multipeak bursts followed by blockade of the LFP. Upon washout, multipeak bursts occur transiently. (**A**) adapted with permission from [59]; (**B**) adapted with permission from [31], copyright (1987) Society for Neuroscience; (**C**) from [20].

**Figure 9 brainsci-12-00437-f009:**
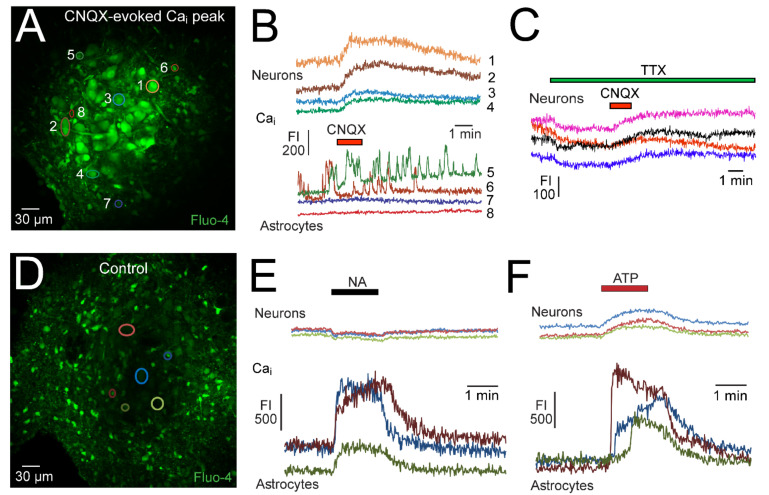
Neuromodulator-evoked changes in the free cytosolic Ca^2+^ concentration (Ca_i_) in the LC of newborn rat slices. (**A**) Fluorescence image (at the peak of the response to bath-applied CNQX) of LC cells bulk-loaded via focal pressure injection with the membrane-permeant form of the green fluorescent Ca^2+^ dye Fluo-4. The numbered colored shapes are regions of interest (ROIs) drawn offline via Fluoview software (FV10-ASW, version 03.01.01.09, Olympus, Markham, ON, Canada) around 4 neurons (# 1-4) and 4 presumptive astrocytes (# 5-8). (**B**) CNQX-evoked Ca_i_ rises, indicated by an increase in Fluo-4 fluorescence intensity (FI), are similar in all 4 ROI-identified neurons whereas the 4 astrocytes labeled do not respond, however 2 cells show spontaneous Ca_i_ rises. (**C**) In 4 neurons of a different slice, the CNQX-evoked Ca_i_ rise persists after preincubation in TTX which decreases Ca_i_ baseline. (**D**) Fluo-4 image with ROIs from 3 neurons and 3 smaller astrocytes (**E**) Ca_i_ kinetics traces from cells in (**D**) plotted during bath-application of 25 µM NA. NA caused a decrease in neuronal Ca_i_ baseline, contrary to eliciting a concentric Ca_i_ wave firstly in peripheral and finally in LC astrocytes (see Appendix A). (**F**) Ca_i_ response of the same cells to bath-application of 100 µM adenosine-triphosphate (ATP). (**A**–**C**) Adapted with permission from [35]; (**D**–**F**) from [20].

**Figure 10 brainsci-12-00437-f010:**
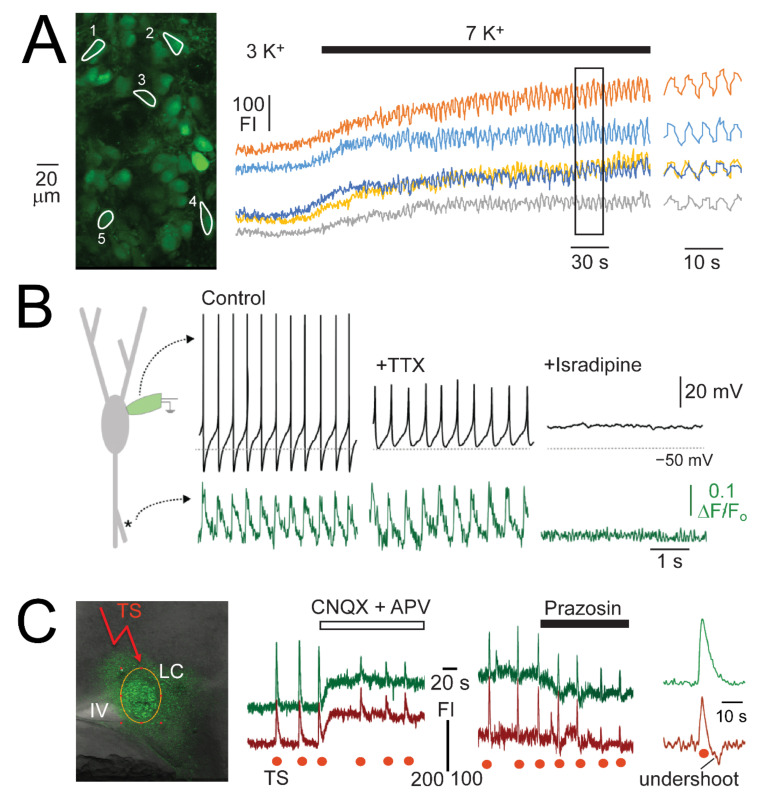
Ca_i_ rises in LC neurons of newborn rat slices during single spikes, K^+^-evoked LFP bursts and electrical stimulation. (**A**) Left, a fluorescence image of LC cells bulk loaded with Fluo-4 with ROI outlines drawn around 5 neurons. The middle trace shows that when superfusate K^+^ is changed from physiological 3 mM K^+^ to 7 mM, baseline FI increases and begins to rhythmically oscillate. These oscillations are shown at a magnified timescale on the right, revealing that they are synchronous. (**B**) Shows a schema of an LC neuron (**left**) during whole-cell V_m_ recording from the soma and synchronized dendritic Ca^2+^ line-scan imaging. Combined voltage (**upper**) and fluorescence (**lower**) traces show that under control conditions and during TTX application dendritic Ca^2+^ oscillations recorded up to 100 μm away from the soma are phase-locked to somatic spiking activity. L-type Ca^2+^ channel antagonist isradipine (1 μM) eliminated Ca^2+^ oscillations in the presence of TTX. (**C**) Shows the LC bulk loaded with Fluo-4 during repetitive electrical stimulation (TS) in an area next to the LC. This resulted in fluorescence increases in neurons (middle traces) that are apparently partly mediated by AMPAR/KAR and ɑ_1_R as they are partly blocked by CNQX and prazosin, respectively. The right traces exemplify that stimulation induced either an increase in FI followed by a return to baseline or FI increases followed by an undershoot below baseline. (**A**) Adapted with permission from [35]; (**B**) adapted with permission from [46]; (**C**) from [20].

## Data Availability

Unpublished data will be made available upon reasonable request to the corresponding author.

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
