# Peer review of "Autocrine Neuromodulation and Network Activity Patterns in the Locus Coeruleus of Newborn Rat Slices"

_brainsci, 2022, doi:10.3390/brainsci12040437_

Round 1

Reviewer 1 Report

This is a nicely written perspective article that summarizes prior research as well as some new observations related to locus coeruleus physiological properties in the neonatal rat.  The authors report that even in the neonatal period, this widely-projecting system that is regulated by neurochemically and neuroanatomically diverse afferents is already quite complex.  I have only minor concerns with the manuscript in its current form, listed below.

First, although the article is quite nicely written, I believe that it could be improved by having it reviewed and edited by a native English speaker

Line 59:  authors state that 5HT is a potential co-transmitter released by LC axons but I am unaware of any such evidence; a reference should be provided

Line 83:  the word somatotopic is not quite the right choice as it implies mapping of physical body spaces onto neural substrates (as in the somatotopic map in primary somatomotor cortices); I think topographical would be a better choice

The language in figure 3 legend is a little confusing as it reports that extracellular spiking is detected with a cell attached electrode.  Similar language occurs in the text around line 208.  Some clarifying explanation of the methods used here would be helpful as I wouldn’t consider cell-attached recording with a glass patch electrode to be extracellular recording.

The data presented in figure 4 showing distinct types of neurons based on their responses to current injection is quite interesting.  Do LC neurons at later developmental time points show similar response properties?  In my experience almost all LC neurons from adolescent and adult rats are of the pacemaker type – some commentary on the differences between neonatal and adult animals in this regard might be useful and thought provoking.

One minor concern is that the figures are not referred to for the first time in sequential order in the text.  For example, Figure 2 is referenced on line 71 before figure 1 is referenced.  This occurs in other locations as well.  It might make more logical sense to present the figures in the text in the order in which they will actually be referenced to.

Reviewer 2 Report

The present study has focused on the neurophysiological features of network activity in Locus coeruleus neurons of neonatal Rats. The subject is interesting for the researchers of the field, results are clear and manuscript is rather well prepared. However, there remains some important points to be addressed/clarified by the authors prior to publication.

Major comments:

  • Recent studies have revealed mutual interactions between orexin and Mu opioid receptors. In addition, the PGi to LC (known as paragiganto-coerulear) connections have widely been investigated in the context of orexinergic-opioidergic systems (see PMIDs: 26230639, 26210888 and 24211689). Thus, these items should be discussed in the manuscript and Fig.1. could be further enriched by adding more mechanistic details in this regard.
  • I did not find the Methods section in the manuscript! This is a very critical matter that must be provided.
  • 2. Stereotaxic coordinates of the recording site should be provided for part B.
  • 7. part C: How do the authors interpret the two downward deflections occurring in current clamp mode?
  • Did the authors measure changes of access resistance (Ra) during intracellular recording throughout their experiments? If yes, it is suggested to mention in methods section and provided data somewhere in Results. This is critical to ensure the stability and reliability of the recorded currents.
  • 8A: the baseline recording does not seem to be stable prior to morphine application. What criteria has been considered in this respect?
  • I expected to see a test pulse at the beginning of intracellular recording sample traces. This is essential to ensure the membrane integrity in whole cell patch clamp mode. As a matter of fact, test pulse should be repeated with reasonable intervals during the experiment.
  • Ethical considerations (including the institutional ethics number issued for this study) must be provided in the manuscript.
  • The total number of animals used and number of cells recorded per slice/rat must be provided.
